# Whole-genome sequencing revealed genetic diversity and selection of Guangxi indigenous chickens

Junli Sun[1‡], Tao Chen[2‡], Min Zhu[1‡], Ran Wang [2], Yingfei Huang[1], Qiang Wei[2], Manman Yang [3]*, Yuying Liao[4]*

1 Guangxi Key Laboratory of Livestock Genetic Improvement, Animal Husbandry Research Institute of Guangxi Zhuang Autonomous Region, Nanning, Guangxi, China, 2 BGI Institute of Applied Agriculture, BGI-Shenzhen, Shenzhen, China, 3 BGI-Shenzhen, Shenzhen, China, 4 Guangxi Veterinary Research Institute, Nanning, Guangxi, China

‡ These authors contributed equally to this work as co-first authors.
* yangmanman@genomics.cn (MY); liaoyuying@126.com (YL)

**Data Availability Statement:** The raw reads data have been submitted to NCBI Sequence Read Archive database with the accession number PRJNA659069. The data reported in this study are also available in the CNGB Nucleotide Sequence

## Abstract

Guangxi chickens play a crucial role in promoting the high-quality development of the broiler industry in China, but their value and potential are yet to be discovered. To determine the genetic diversity and population structure of Guangxi indigenous chicken, we analyzed the whole genomes of 185 chickens from 8 phenotypically and geographically representative Guangxi chicken breeds, together with 12 RJFt, 12 BRA and 12 WL genomes available from previous studies. Calculation of heterozygosity (Hp), nucleotide diversity (π), and LD level indicated that Guangxi populations were characterized by higher genetic diversity and lower differentiation than RJFt and commercial breeds except for HGFC. Population structure analysis also confirmed the introgression from commercial broiler breeds. Each population clustered together while the overall differentiation was slight. MA has the richest genetic diversity among all varieties. Selective sweep analysis revealed *BCO2*, *EDN3* and other candidate genes had received strong selection in local breeds. These also provided novel breeding visual and data basis for future breeding.

## Introduction

Chickens are the most widely distributed livestock species globally; more than half the total (53%) is found in Asia, one of the largest producers in China [1]. In China, poultry meat consumption accounts for the second-largest proportion after pork. People in different regions have different preferences for the appearance, flavor, and cooking methods of chickens.

Guangxi Zhuang Autonomous Region is in mountainous terrain in the far south of China, and its unique climate has created unique and rich chicken germplasm resources. Three-yellow chicken (SHC) with yellow feathers, skin and shank, is a favorite choice for traditional broths and soups in southern China. Nandanyao chickens (NDYC) are famous for lower fat deposition and better meat quality [2]. Longshengfeng chickens (LSFC) have compact bodies,

Archive (CNSA: https://db.cngb.org/cnsa; accession number CNP0001716). Furthermore, 36 individuals of commercial breeds and red jungle fowls were downloaded from NCBI at ERP112703 (S2 Table).

**Funding:** Yy L, grant number: AA17204024, Guangxi Special Project for innovation-driven development, Yy L played a role in study design and decision to publish. Mm Y, grant number: JCYJ20180307163440037. Shenzhen Municipal Government of China, Mm Y played a role in preparation of the manuscript.

**Competing interests:** The authors have declared that no competing interests exist.

feathered legs, and various feather color patterns [3]. Xiayan chickens (XYC) and Guangxima chickens (MA) are characterized by large size and tender meat and are known for a dish "white sliced chicken". Guangxiwu chicken (WC) has black feathers, bones and meat. Dongzhongai chicken (DZAC) and Cenxigudian chicken (GDC) are two characteristic populations with a long local breeding history.

Previously studies on Guangxi chickens focus on growth performance, meat quality, and feed efficiency [2, 4, 5], or their genetic diversity using low-density markers. Liao et al. assessed the genetic diversity of Guangxi chicken breeds with 18 microsatellite loci and the mitochondrial DNA D-loop region [6]. Yang et al. performed an analysis of the genetic difference of Guangxi native chicken and no associated genes were prominent might duo to the deficient of RAD-seq and grouping [7].

The approach of whole-genome re-sequencing (WGRS) has proven to be a powerful tool for genetic evaluation, selective sweep analysis and genetic relationship exploration. Huang et al. identified *BCO2*, *RALY*, *lLGR4*, *SLC23A2*, and *SLC2A14* as candidates for pigment determining genes by genome-wide scans in Three-yellow chickens [8]. Huang et al. have uncovered the genetic structure and the molecular underpinnings of the SHCs trademark coloration using WGRS data [8]. Li et al. have explored the genetic signatures of high-altitude adaptation in Tibetan chickens by comparing the strong selection signatures genomic region of the Tibetan and lowland fowls [9]. Luo et al. performed a comparative genomics analysis for determining the behavioral pattern of gamecock chickens and observed genetic introgression from commercial chickens into indigenous chickens [10].

A comprehensive and deep understanding of the genome information of the indigenous breeds could reveal the genetic diversity and population structure of these breeds. This study, therefore, investigated genetic diversity, population structure, linkage disequilibrium (LD), and signature selection within Guangxi indigenous chickens using genome-wide single nucleotide polymorphisms (SNPs) generated from the whole genome sequencing.

## Materials and methods

### Ethics statement

This study was carried out following the Animal Experimental Ethical Inspection Form guidelines of Guangxi Research Institute (20190318).

### Sampling and genotyping

A total of 185 blood samples from six breeds and two characteristic populations were investigated from conservation centers, or breeding farms (S1 Table and S1 Fig) were collected for genomic DNA extracting. The MGIEasy Universal DNA Library Prep Set constructed Whole-genome sequencing libraries and then sequenced using MGISEQ-2000 with PE100 developed by BGI Genomics Co., Ltd.

### Variant calling and annotation

After quality control, Pair-end reads were mapped onto the *Gallus gallus* GRCg6a (https://www.ncbi.nlm.nih.gov/assembly/GCF_000002315.6/) using BWA version 0.7.12-r1039 [11]. The bam files were sorted using SortSam and duplicated reads were marked using MarkDuplicates from Picard tools version 1.105. SNPs were detected and filtered using HaplotypeCaller and VariantFiltration command in GATK version 4.1.1.0. We applied hard filter command 'VariantFiltration' to exclude potential false-positive variant calls with the default parameter "QD < 2.0 || ReadPosRankSum < -8.0 || FS > 60.0 || MQ < 40.0 || SOR > 3.0 || MQRankSum

< -12.5 || QUAL < 30". To annotate the SNPs and InDels identified here, Vep (v95.3) was employed (http://asia.ensembl.org/info/docs/tools/vep/index.html).

### Genomic diversity analysis

Genome-wide nucleotide diversity ($\pi$) and genetic differentiation (Fst) was performed using VCFtools (v0.1.13) [12] with parameters 40kb sliding window and 20kb step size. Individual heterozygosity (Hp) was calculated by following the formula given by Rubin et al. [13]:

$$Hp = \frac{2 \sum n_{MAJ} \sum n_{MIN}}{\left(\sum n_{MAJ} + \sum n_{MIN}\right)^2}$$

PopLDdecay version 3.40 [14] was used to assess patterns in the extent of linkage disequilibrium. ROH (runs of homozygosity) analysis was performed using plink version 1.9, the parameters were as follows, the minimum length of ROH > 10kb, the number of SNP per window > 20, and only one heterozygote were allowed, ROH had at least one variant per 1000 kb on average. The results of ROH analysis were visualized with the R package pheatmap, the total ROHs length of each chromosome was centered and scaled in breed's level.

### Population structure analysis

To investigate the genetic background of the chickens, principal component analysis (PCA) and structure analysis were conducted. SNPs in high linkage disequilibrium were removed by PLINK version 1.9 [15]. The pruned SNP data estimated the individual ancestries using a maximum likelihood method implemented in the ADMIXTURE version 1.23 [16]. The default parameter (folds = 5) for cross-validation and the lowest cross-validation error was taken as the most probable K value. VCF2Dis (v1.09) software was used to calculate the P distance matrix, then use PHYLIPNEW (v3.69) to construct an NJ-tree.

### Sweep analysis, GO enrichment and KEGG pathway analysis

The selective sweep analysis was performed using vcftools (v0.1.13) in Guangxi indigenous chickens. Scanning the whole genome selection signal with 40kb as the sliding window and 20kb as the step size and windows with less than 10 SNPs are excluded [17]. Fst values were Z-transformed: $ZFst = \frac{(Fst-uFst)}{\sigma Fst}$, where $\mu$ is the mean of Fst and $\sigma$ is the standard deviation of Fst. The sliding window with the top 1% of the ZFst value was defined as a significantly selected region [18]. GO enrichment analysis and KEGG pathway analysis were performed using the packages clusterProfiler, KEGG.db and org.Gg.eg.db in R. We select Benjamini-Hochberg method correction for multiple comparisons and GO terms with a p-value less than 0.05 were considered significantly enriched.

## Results

### Variant calling and annotation

A total of 9.48 billion clean reads were obtained after quality filtering, corresponding to average depth and coverage is 9.39x and 96.97% (S3 Table). The overall mapping rate is greater than 98.4%. SNPs with MAF < 0.05, call rate < 0.8 and individual call rate < 0.9 were excluded, 13,245,769 high-quality SNPs and 3,790,305 indels were utilized for downstream analysis (Fig 1A). The Guangxi indigenous chickens harbored a higher number of SNPs and indels than that of RJFt except for DZAC, while WL exhibited the lowest ones. Vep was used to annotate SNPs, 55.66% of these SNPs were aligned to the intron region, 28.43% were aligned

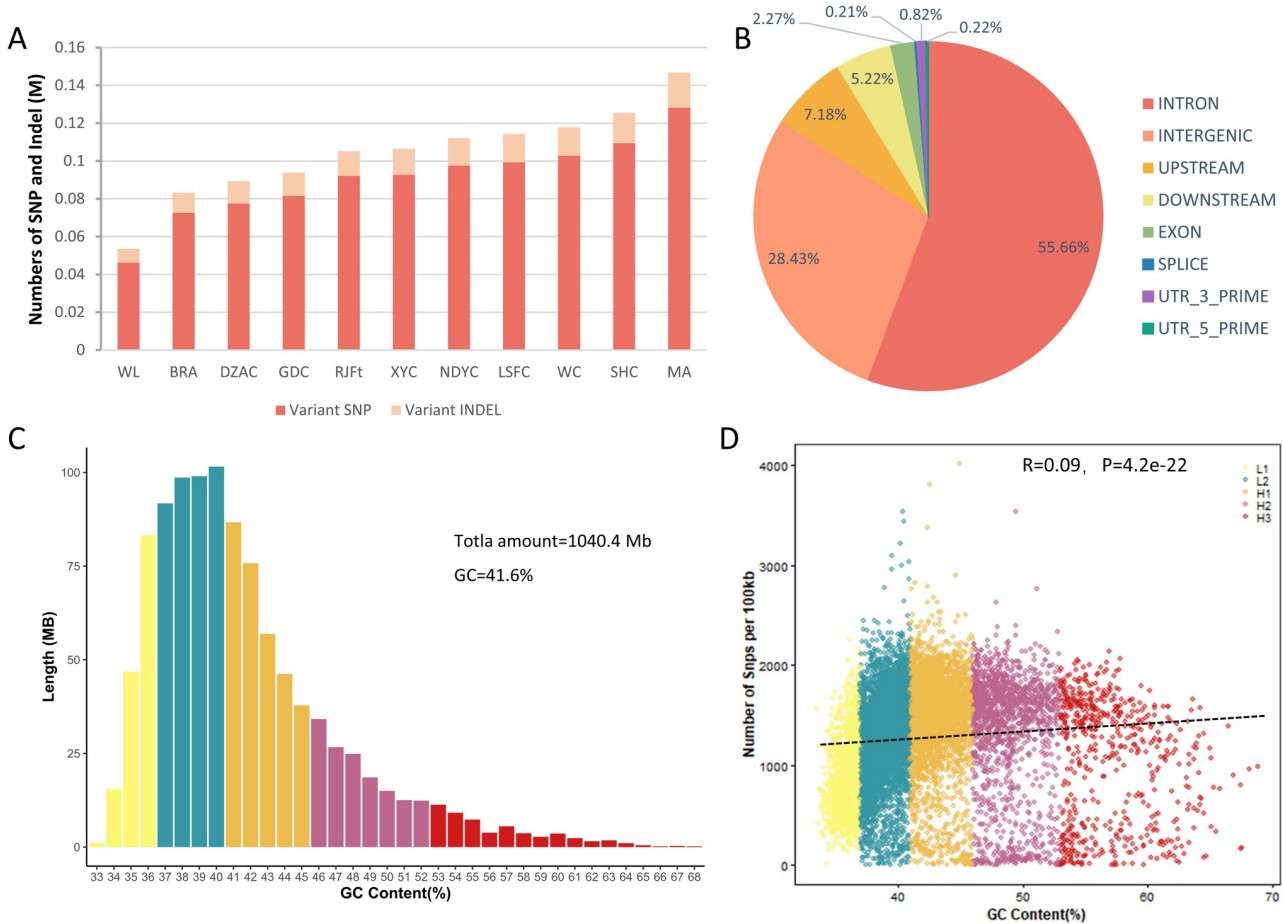

**Fig 1. The number, distribution, and GC content of SNP and INDEL.** (A) The SNP and indel number in different breeds. (B) The distribution of SNPs. (C) Distribution of isochores according to GC levels. (D) Scatter plot of SNP number and GC content in isochrones per 100kb window.

to the intergenic region, and only 2.27% were located in the exon region (Fig 1B). Compared with chicken SNP data from Ensembl database (http://ftp.ensembl.org/pub/release-104/variation/gvf/gallus_gallus/), 1,280,234 were assigned as novel SNPs and included 42,563 SNPs (accounting for 1.57% of the novel SNPs) were located in the coding regions. Among the coding SNPs, there were 27,133 synonymous mutation SNPs (63.75%) and 15,159 missense mutation SNPs (35.62%).

Isochores are long DNA fragments with uniform GC content, are tightly associated with many genomic biological characteristics such as recombination, GC3 content, and gene density [19]. The genome is divided into isochrones with a sliding window of 100kb and divided into five categories (L1, L2, H1, H2 and H3) according to different GC levels to explore the potential impact between GC content and genetic variations [9]. Our results showed that Guangxi chickens have a mosaic structure of isochores, the major isochores are the GC content of 36–42%, comprising a part of L2 and H1, which is the main source of variation. The L2 category has the largest number of isochrones, covering 37% of the genomic region, and the SNPs and Indels count peak in this category (Fig 1C and 1D). H1 category with a higher GC level also contains a lot of genetic variations (S5 Table). In general, genomic regions with moderate GC content contain more variation. We used the Pearson correlation coefficient to calculate

the relationship between GC content and the number of SNPs/indels. The number of SNPs/indels and the GC level are weak positively correlated in the isochores of the chicken genome (r = 0.09, p = 0; r = 0.13, p = 0) (Fig 1D and S3C Fig).

## Genetic diversity, LD and ROH analysis

To provide a more comprehensive understanding and profound insight into the genome diversity of Guangxi indigenous chickens, we incorporated the sequencing data of 12 Red jungle fowl population from Thailand (RJFt) and commercial breeds including 12 white layers (White Leghorn, WL) and 12 Broiler A(BRA), which has been previously published [17]. The nucleotide diversity (π) and heterozygosity were calculated to evaluate the genetic diversity of all the chicken breeds. We observed Guangxi indigenous chickens harbored the higher genome-wide π than RJFt (π = 0.00334) except for DZAC (π = 0.00332), the lowest genome-wide π in WL (π = 0.00152), followed by BRA (π = 0.0031) (Fig 2A). Unlike the results in nucleotide diversity, the heterozygosity level of Guangxi chickens is generally low. DZAC followed by BRA, which harbored the highest heterozygous SNP rate (He = 0.2730), while MA harbored the lowest (He = 0.1684) one (Fig 2B). The He in Z chromosome is lower than any autosome among all populations (S2 Fig), probably because the sex chromosome had undergone higher selective pressure than the autosome [20]. The level of ROH reflects the recent inbreeding history of a population [21]. As shown in Fig 2C, the average and total length of ROH of indigenous chicken are short, RJFt had the lowest number of ROH. The difference of mROH among Guangxi breeds is relatively small (ranging from 37.74 kb to 66.78kb), which is

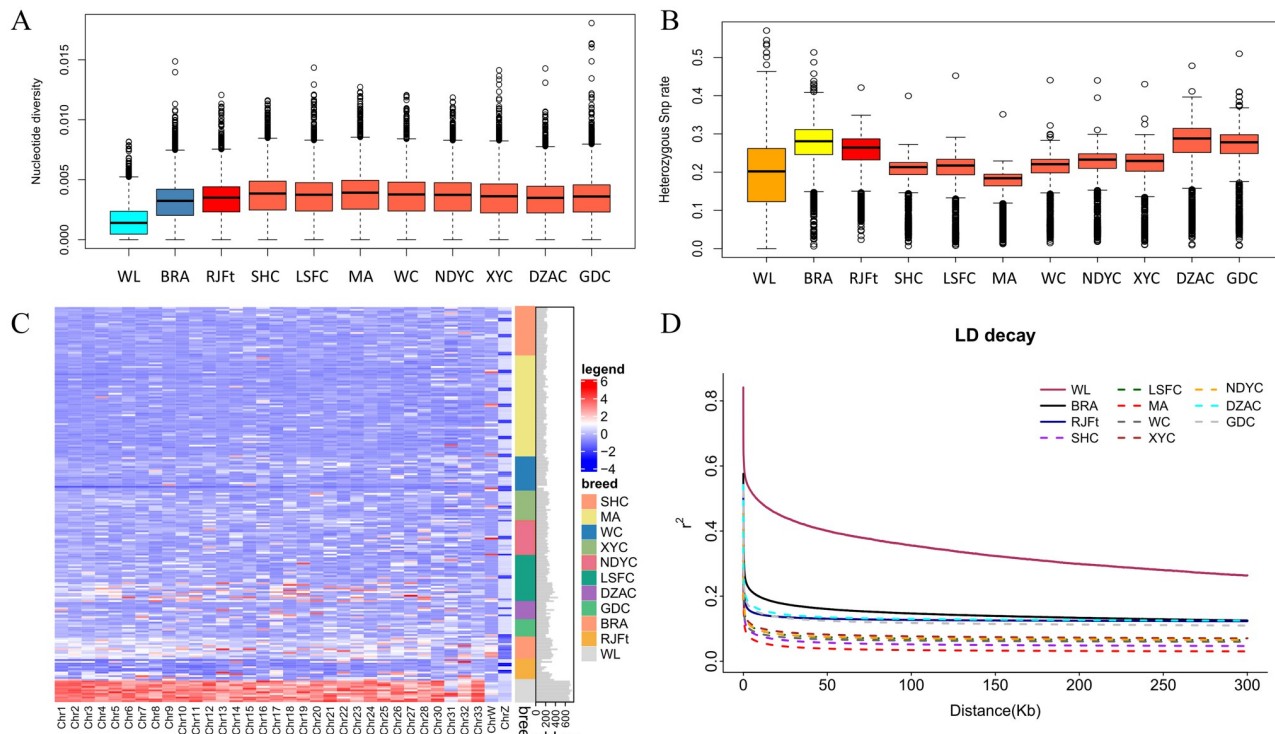

**Fig 2. Genome diversity and LD decay of 11 chicken populations.** (A) Genome nucleotide diversity was calculated with a window size of 40 kb and a step size of 20 kb. (B) Genome heterozygosity within 500kb sliding window across the genome. (C) The ROH of each chromosome in different breeds. The redder color represents longer ROH, the bluer the shorter. (D) Linkage disequilibrium (LD) decay, denoted with one line for each population.

very different from the largest value of 167.3 kb in WL (S6 Table). HGFC (347.039Mb) had the same tROH level as BRA (338.084 Mb), and the shortest tROH was observed in LYWC (185.483 Mb). Linkage disequilibrium (LD) analysis showed that the WL population had the slowest LD decay rate, significantly slower than the followed BRA. MA had a faster LD decay rate than other chicken breeds, DZAC and WC have similar LD levels with RJFt in the second group (Fig 2D).

## Population genetics analysis

As expected, the chickens from the same breeds clustered together according to the PCA. The PC1 (26.79% variances explained totally) could separate the commercial layer breed WL from other populations and PC2 (8.86% variances explained totally) displayed the genetic differentiation between commercial broiler breed BRA and other populations except for HGFC (Fig 3A). The 12 RJFt gathered with Guangxi fowls and away from commercial chickens. When WL, BRA, RJFt and HGFC were removed from the dataset, DZAC and GDC could be identified as separate clusters, the two populations of LSFC, MC and LSFC, are distributed separately. DZAC and LSFC have a significant variation within the breed. MA, SHC and XYC are geographically close and tend to get more relative to each other.

PCA results could not completely reproduce the phylogenetic relationships, and the neighbor-joining tree corroborates the findings of the PCA (Fig 3B). Individuals from the same breed gathered were consistent with the breeding history and geographical distribution. Taking the RJFt as the root, part of the LSFC were clustered with the outgroup background (BRA and WL). Then the rest of LSFC, WC and NDYC are grouped. Following this group, the local populations DZAC, GDC and several MA were arranged in the middle of the tree but were not forming a visually distinct cluster. MA has a wide range of sampling sources within abundant

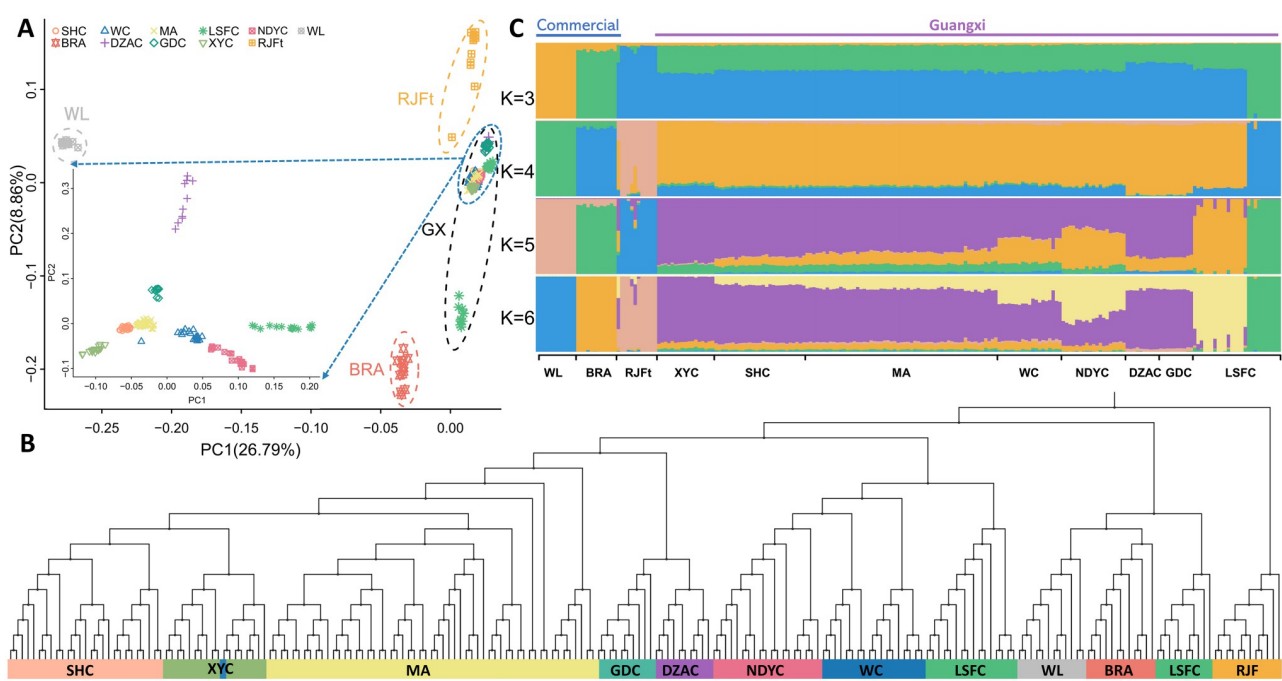

**Fig 3. Population structure analyses.** (A) Principal component analysis (PCA), with 26.79% and 8.86% variance explained in PC1 and PC2, respectively. (B) Neighbor-joining tree of 221 chickens, constructed with PHYLIPNEW version 3.69.650. (C) Admixture analysis with K values running from 3 to 6.

variation and is clustered separately. The SHC and XYC branched into two sub-clusters according to the sampling sources, an individual from the WC was grouped with XYC, possibly because of a sampling error.

To assess historical admixture patterns of the chickens, we conducted the ADMIXTURE analysis with K values running from 3 to 18. At K = 3, genetic divergency first occurred between commercial breeds and non-commercial ones. HGFC shared the same ancestral lineage with BRA, Guangxi indigenous breeds shared the same ancestral lineage with RJFt, WL consistent with the above PCA and phylogenetic tree result (Fig 3C). When K = 4, the Guangxi indigenous breeds were separated from others (except HGFC). Indigenous chickens gradually separated from each other when K ranged from 5 to 14. MA experienced introgression from SHC, and the ancestral components of SHC, GDC and HGFC are pure. There is differentiation in the breeding programs of LYWC and DLWC, as well as in LSFC and XYC breeds (S4 Fig). According to the calculated cross-validation value, the best fit was K = 5, Guangxi indigenous breeds showed two ancestral components that are different from others.

## Selective sweep analysis

The skin color influences consumers' preferences, and yellow skin chickens are more popular than white ones in the south of China. According to the color of chicken skin, we divided the population into yellow skin groups (XYC, SHC, MA, DZAC and GDC) and non-yellow skin groups (LSFC, NDYC and WC). We observed the highest ZFst region occurring at chr24: 6.14–6.18 Mb (Fst = 0.63, ZFst = 35.39) between 121 yellow skin and 64 non-yellow skin chickens by the selective sweep (Fig 4A). The top ten selected window annotated eight genes,

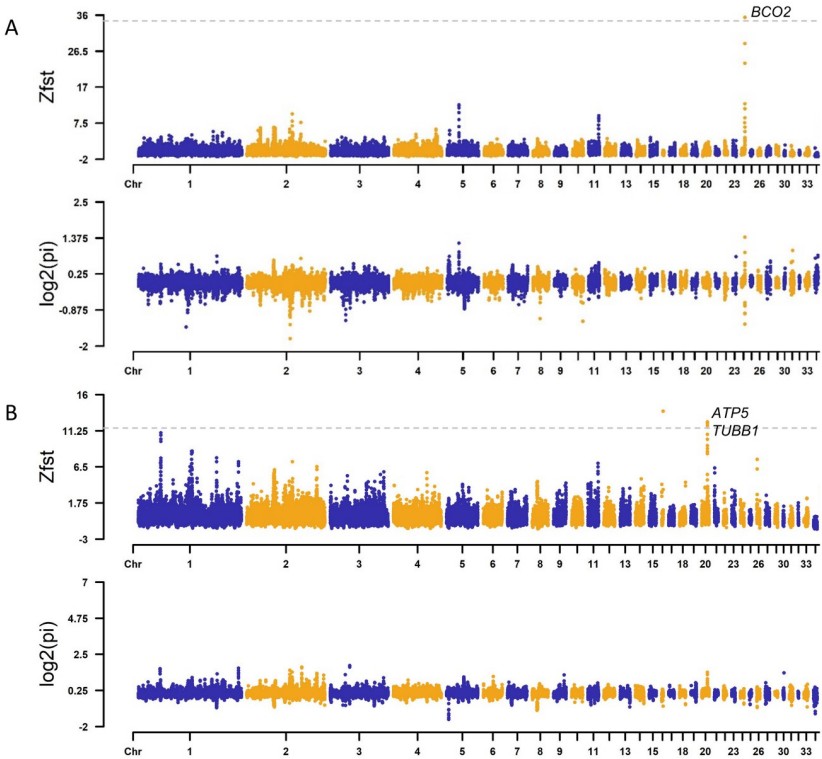

**Fig 4. The result of ZFst and Log2(pi) of Guangxi chicken.** (A) Yellow skin and non-yellow skin chickens. (B) WC and other indigenous breeds. The horizontal dotted lines represent the top 1% cut-off.

among which the *BCO2* gene is a classical yellow color gene in chicken. Then we collected the genotypes on this gene and found that the non-yellow skin clusters showed a different genotypic pattern from LSFC and NDYC; as for WC, it has both two patterns might due to the concealment of its black skin that the yellow skin has not been deliberately selected (S6 Fig). We found a missense mutation at chr24: 6155481T>C (rs313409504) was consistent with the previous report [22]. The strongest selective sweep on chromosome 11 we detected was located at 19.12–19.16Mb (Fst = 0.17, ZFst = 9.41) near the gene *MC1R*, which plays a key role in controlling the deposition of melanin.

In oriental countries, nutritional and medicinal benefits have been attributed to the consumption of black-boned chickens. WC is distinguished from other breeds because of its black beak, crown, skin, and shank. We compared WC with other Guangxi chickens and scanned the whole genome for selected regions (Fig 4B). The strongly selected region contained four annotated functional genes which were associated with dermal hyperpigmentation in chickens, beta-1 tubulin (*TUBB1*) and *PRELI* domain containing 3B (*PRELID3B*, also known as *SLMO2*), *GNAS* complex locus (*GNAS*), encoding endothelin 3 (*EDN3*). GO enrichment analysis shows that gene *TYRP1* on chromosome Z and *KITLG* on chromosome 1 were significantly enriched in melanocyte differentiation (GO:0030318) and developmental pigmentation (GO:0048066). *SYK* gene plays a role in the regulation of bone resorption (GO:0045124).

XYC and GDC have a similar appearance to SHC, XYC is fat-deposited and GDC is smaller. We compared them with the SHC and scanned the whole genome for selected regions related to fat deposition and body size. We found that the selected genes of fat deposits were located on chromosome 12 and chromosome Z. *HMGCS1* and *OXCT1* are significantly enriched in the pathway of ketone body synthesis and degradation (gga00072), ketone bodies are produced in the liver, mainly from the oxidation of fatty acids, and are exported to peripheral tissues for use as an energy source. *ATG7* (Autophagy Related 7) is a protein-coding gene (gga:04140), it has been associated with multiple functions, including axon membrane trafficking, axonal homeostasis, mitophagy, adipose differentiation, and hematopoietic stem cell maintenance. A significant signal peak was detected at 49.32–49.36 Mb on chromosome 5 (ZFst = 8.5, Fst = 0.51) located in the *DLK1-DIO3* genomic region, which was recognized to be an imprinted domain in placental mammals associated with developmental programming [23].

## Discussion

We assessed the genetic diversity of indigenous chickens from Guangxi provinces using the resequencing. Meanwhile, we performed a selective sweep analysis of phenotypes related to economic traits.

Compared with the RAD-seq [7] in Guangxi chicken, we obtained dozens of times more SNPs than them through WGS, especially in Ma, perhaps because our samples came from multiple populations. About 1.2Mb are novel SNPs, which is consistent with the previous reports in local chickens. Of these novel SNP, 1.57% are located in the coding region, including 186 stop-gain and 16 stop-loss SNP, some nonsense mutations are functional important [13]. The results of isochores analysis showed that L2 and H1 were the dominant sources of genetic variation, and the gene density was higher in the GC-rich region (S3A and S3B Fig). Compared with the previous research in Sichuan chickens [9], the single nucleotide mutation rate in Guangxi chicken has a weak positive correlation with the genome's GC content, which is probably because we employed different filtering methods to retain more variation in the GC-poor region. Guangxi chicken has a relatively high level of genetic diversity ($\pi > 0.003$), and most breeds even higher than RJF, it could be due to the low degree of intensification and have not

undergone intensive selection as experienced in other breeds(e.g. White Leghorn), or may be related to the excessive concentration of RJF sampling [17]. In contrast, the heterozygosity level of Guangxi chicken is low, especially in MA.

The different patterns of ROH length can explain the difference between breed origin and recent management [24]. Guangxi chickens have a higher proportion of ROH in shorter ROH categories except for LSFC, which indicates that a small number of founding populations may have initially established them but were not be highly affected by recent inbreeding [25], this is consistent with the results of structural analysis. In PCA analysis, Guangxi chickens were neither close to broilers nor layers but gathered alone near RJFts. However, Guangxi chickens may have been selected for meat in the breeding process, which can be seen in the structure of Guangxi chickens mixed with the ancient composition of BRA. HGFC, as a population of LSFC breed, has detected gene introgression from commercial broilers. After further investigation, to improve the economic benefit of this group, broilers genes were introduced artificially in the process of breeding. The results of the genetic structure are consistent with the origin of breeds, indicating the effectiveness of Guangxi's local chicken population in breed protection, even the HGFC also formed its characteristics because the ancestral composition of it is different from that of BRA when k = 6. Compared with commercial breeds, the genetic difference among indigenous chickens is relatively small, and Guangxi indigenous fowls have a closer relationship with RJFt [26].

The difference in deposition location and amount of carotenoid and melanin in chicken skin led to the diversity of skin color. *BCO2* gene encodes beta-carotene dioxygenase 2 could cleave colorful carotenoids to colorless apocarotenoids by an asymmetric cleavage reaction [27], is established as the causal gene for the yellow skin. According to the results of selection scanning, the *BCO2* gene is extremely strongly selected in the population, and the SNP shows different patterns in the yellow and non-yellow skin populations. Eriksson et al. demonstrate that regulatory mutations that inhibit expression of *BCDO2* in the skin caused yellow skin, but not in other tissues [22]. Fallahshahroudi's study showed the down-regulation of *BCO2* in skin, muscle, and adipose tissue was associated with the derived haplotype [28]. Also, *BCO2* has various variants in different breeds. Wang found a G>A mutation in exon 6 to be associated with the concentration of carotenoids in Guangxi-Huang and Qingjiao-ma chicken [29]. A GAG haplotype was fixed in commercial breeds of yellow skin [22]. We also found the missense mutant at chr24:6155481 led to the mutation of threonine to alanine.

The strongest selective sweep region with dermal hyperpigmentation on chromosome 20 was located at 10.64–10.94 Mb consisting of seven genes. *EDN3* is a gene with a known role in promoting melanoblast proliferation by encoding a potent mitogen for melanoblasts/melanocytes. Shinomiya et al. reported that the overexpression of genes in a 130kb duplication region gives rise to the hyperpigmentation in silk chickens [29], and then Dorshorst et al. extended this discovery and strongly suggested that the increase of *EDN3* expression caused by duplication is the cause of *FM* in all breeds of chickens, the expression of two other genes, *SLMO2*, and *TUBB1* were also significantly increased in expression in both skin and muscle tissue from adult fibromelanosis chickens might contribute to the dermal hyperpigmentation phenotype [30]. Analysis of RNA-seq suggested that *SLMO2*, *ATP5e*, and *EDN3* were differentially expressed between the black and yellow skin groups, combined analysis of genomic data found that *EDN3* might interact with the upstream ncRNA *LOC101747896* to generate black skin color during melanogenesis [31]. Wang et al. study indicated that a T2270C mutation in *GNAS* gene promoter in chicken is correlated strongly with the skin color traits [32].

*DLK1* has been reported to be associated with the development and differentiation of adipose and muscle in chicken and *DLK1* promotes muscle development inhibitory adipogenesis in mammals [33]. Zhang et al.'s study provides strong in vivo evidence that atg7, and by

inference autophagy, is critical for normal adipogenesis [34]. *AMACR* coding protein alpha-methylacyl-CoA racemase is involved in the pathway bile acid biosynthesis, which is part of Lipid metabolism (gga00120). Bile acid is the main component of bile and its main function is to promote the digestion and absorption of fat. *HSD17B4* codes a bifunctional enzyme mediating dehydrogenation and hydration during β-oxidation of long-chain fatty acids, and a non-synonymous SNP has been reported to be related to meat quality traits in pigs [35]. *PRKAA1* is associated with skeletal muscle lipid accumulation [36].

## Conclusions

In conclusion, a comprehensive analysis of genetic diversity, population genetic structure, LD, and the selection signatures of 8 indigenous chicken breeds distributed in Guangxi. The results suggested that indigenous chickens have abundant genetic diversity and potential, candidate genes related to economic traits can also provide a theoretical basis for breeding. Our analyses provide data for further research and local breeding of Guangxi indigenous chicken.

## Supporting information

**S1 Fig. Geographic distribution and appearances of typical female chickens.**
(DOCX)

**S2 Fig. Boxplot showing heterozygous SNP rate of autosomes (left) and Z chromosome (right) between each chicken population.**
(DOCX)

**S3 Fig. Isochores and ROH distribution of Guangxi chicken.** (A) Distribution of isochron along chromosome. (B) Chromosome length and gene density. (C) Scatter plot of Indel number and GC content in isochrones per 100kb window. (D) Distribution of ROH in chicken breeds.
(DOCX)

**S4 Fig. Admixture analysis with K values running from 7 to 18.** Each population separated by white dotted line.
(DOCX)

**S5 Fig. The genotype of fixed SNPs in chr24: 6.14Mb~6.18Mb of individuals.** The row represents the SNP position and the column represents the individual. Light blue denotes reference alleles while red indicates alternative homozygous alleles, yellow means heterozygous and dark blue means missing.
(DOCX)

**S6 Fig. ZFst values and Log 2 (pi).** (A) XYC and SHC. (B) GDC and SHC.
(DOCX)

**S1 Table. The character of chickens in this study.**
(XLSX)

**S2 Table. The public data information.**
(XLSX)

**S3 Table. The sequencing information of samples.**
(XLSX)

**S4 Table. The distribution of variant, nucleotide diversity and HE.**
(XLSX)

**S5 Table. The distribution of isochrones.**
(XLSX)

**S6 Table. ROH of chicken breeds.**
(XLSX)

**S7 Table. Selective sweep of dermal hyperpigmentation.**
(XLSX)

**S8 Table. Selective sweep of yellow skin.**
(XLSX)

**S9 Table. Selective sweep of body size.**
(XLSX)

**S10 Table. Selective sweep of fat deposition.**
(XLSX)

**S11 Table. Functional gene categories enriched for genes under selection.**
(XLSX)

**S12 Table. KEGG pathway analysis of genes.**
(XLSX)

## Acknowledgments

We would like to thank the members of Guangxi key laboratory of livestock genetic improvement, BGI Institute of Applied Agriculture, BGI-Shenzhen, and Guangxi veterinary research institute for helpful input on the project.

## Author Contributions

**Conceptualization:** Min Zhu, Yuying Liao.

**Data curation:** Tao Chen.

**Formal analysis:** Junli Sun, Min Zhu.

**Funding acquisition:** Yuying Liao.

**Investigation:** Junli Sun, Yingfei Huang.

**Methodology:** Ran Wang, Qiang Wei.

**Project administration:** Ran Wang, Qiang Wei.

**Resources:** Yingfei Huang, Yuying Liao.

**Visualization:** Tao Chen.

**Writing – original draft:** Junli Sun, Tao Chen.

**Writing – review & editing:** Manman Yang.

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
