## [Decision Letter · Decision Letter 0]

22 Jul 2021

PONE-D-21-10366

Whole genome sequencing revealed genetic diversity and selection of Guangxi indigenous chickens

PLOS ONE

Dear Dr. Yang,

Thank you for submitting your manuscript to PLOS ONE. After careful consideration, we feel that it has merit but does not fully meet PLOS ONE’s publication criteria as it currently stands. Therefore, we invite you to submit a revised version of the manuscript that addresses the points raised during the review process.

We look forward to receiving your revised manuscript.

Kind regards,

Tzen-Yuh Chiang

Academic Editor

PLOS ONE

Journal Requirements:

Reviewers' comments:

Reviewer's Responses to Questions

**Comments to the Author**

1. Is the manuscript technically sound, and do the data support the conclusions?

Reviewer #1: Yes

Reviewer #2: Yes

2. Has the statistical analysis been performed appropriately and rigorously? 

Reviewer #1: Yes

Reviewer #2: N/A

3. Have the authors made all data underlying the findings in their manuscript fully available?

Reviewer #1: Yes

Reviewer #2: Yes

4. Is the manuscript presented in an intelligible fashion and written in standard English?

Reviewer #1: Yes

Reviewer #2: Yes

5. Review Comments to the Author

Reviewer #1: This is a regular experiment but we also find something interesting point in this paper. Atfirst, the author analyzed the genetic diversity and population structure of Guangxi indigenous chicken based on the whole genome, this is a good point. Furthermore, selective sweep analysis is the highlights for this paper. But before published on Plos One, there are some mistakes need to be corrected. The major comments are as the following while minorpoints have been marked in the PDF file.

1. Introduction. This part I don’t think the author introduced the background of this paper well. It’s only something about the situation of the chickens, but the genetic diversity and why the author performed the selective sweep analysis is unknown.

2. M&M. line 68-70: the author provided some paraments for variant calls, but what is the basis for using these parameters?

3. Results. In line 114-117: The authors have divided the data into five categories, what is the significance of the existence of the categories? No subsequent analysis was seen.

4. Results. There are many changes to be made to the images. Fig3(A): The number of samples presented in the graph does not match the graph notes. There are many samples not shown in the figure notes. Fig3(B): BRA and DZAC colour distinction is not very good. It is difficult to see the difference between the two in the samples. Fig3(C): The result to be presented in the diagram is not clear.

Three diagrams of similar samples in as close a colour match as possible

5. Results. The principle of Fig. 4 is not made clear. What analysis is Figure 4 based on to obtain the results?

6. Discussion. No good explanation of the results of the genetic diversity please revised it clearly and find for materials to enrich your discussion.

Reviewer #2: My comments to authors are below (not in any particular order). The paper is brief, but very interesting, and deserves publication in Plos One after a few important corrections.

- I found the introduction very short and lacking information. For instance, it is very important for the reader to know what other efforts have been made to caracterize the Guangxi breeds? Have they ever been sequenced? Has similar work been done? What has not been done? What could justify the use of WGS over genotyping? Have the breeds been genotyped before? If they are in an official germplast catalog, they must have? Such studies are alluded to at lines 44 and 45 when "sporadic studies" are mentioned but we do not have any details. Please detail these studies, as this will make your study even more justified, which is important for the reader to contextualise your study with what has been done with these particular breeds, and also with other competing indigenous breeds in China.

- Also in the introduction, sentences like "Indigenous chickens with delicious meat quality and unique flavors are more in line with the habits and preferences of domestic consumers" (lines 34-35) are highly subjective and a bit ideological. Has there been any study of choice preference or purchasing habits in this region to support this? I understand the need to breed from diverse stocks and especially from indigenous stocks, but it was not made apparent in this section.

- Regarding variant calling: did author estimate how much of the indigenous breed genomes did not map to the GRCg6a genome because of divergence? I understand it is difficult to proceed without reference genomes but you have sequenced the whole genomes. From an agricultural and breeding perspective, it would be quite interesting to know how much of the indigenous genomes is not represented by reference genomes from other breeds, and potentially how much "genetic novelty" there is in those genomes. For instance, it could be interesting to see this on Figure 1A (how much % instead of number of bases).

- Regarding figures, Figure 1A lacks appropriate Y-axis legends. Please correct. Generally speaking, all figures of this manuscript are quite low-quality (both in execution and in resolution). However, one thing that would be beneficial is to increase the font size. Many writings are not readable (especially in Figure 3A). It would be good to keep similar/related colour code for Figure 2D (I found myself struggling to remember the breed names I am unfamiliar with, perhaps worth providing the full names or whether they are Guangxi or not in the legend/color code too).

- I am not a fan at all of the radial tree for the representation of diversity in Figure 3B. I also think the color code of this panel is very hard to read. It would be better to show a rectangular tree, rooted on red-jungle fowls and annotated clearly with all the breeds rather than showing it like this.

- I suspect WL is White Leghorn, and not White Layers. Please correct in the manuscript where applicable.

- Please annotate Figure 4 with the loci of interest and thresholds to consider an interesting locus or not. It would be much more useful than the current version.

6. PLOS authors have the option to publish the peer review history of their article (what does this mean?). If published, this will include your full peer review and any attached files.

Reviewer #1: No

Reviewer #2: No

---

## [Author Response · Author response to Decision Letter 0]

13 Oct 2021

Dear Chiang and dear reviewers, 

Re: Manuscript ID: PONE-D-21-10366 and Title: Whole genome sequencing revealed genetic diversity and selection of Guangxi indigenous chickens.

Thank you for your letter and the valuable comments from you, the assistant editor and reviewers. Those comments were highly insightful and enabled us to improve the quality of our manuscript significantly. We have studied the comments carefully and tried our best to revise the manuscript. The point to the point response to the reviewer's comments are listed as following: 

Revisions in the text are shown using the yellow highlight for additions and strikethrough font [example] for deletions.

Reviewer #1: The major comments are as the following while minorpoints have been marked in the PDF file.

1. Introduction. This part I don't think the author introduced the background of this paper well. It's only something about the situation of the chickens, but the genetic diversity and why the author performed the selective sweep analysis is unknown.

Response: Thank you your valuable advice. The introduction has supplemented the necessity of conducting genetic diversity and selective sweep analysis in LINE 38-62.

Here, I briefly describe: 

Previously studies on Guangxi chickens primarily focus on growth performance, meat quality, and feed efficiency (Zhou and Wang 2011; Du W et al. 2020; Xiao et al. 2021), or their genetic diversity using low-density markers. Liao et al. assessed the genetic diversity of Guangxi chicken breeds with 18 microsatellite loci and the mitochondrial DNA D-loop region (Liao et al. 2016). Yang et al. performed an analysis of genetic differences of Guangxi native chicken and no associated genes were prominent might duo to the deficient of RAD-seq and grouping (Yang et al. 2020). The approach of whole-genome re-sequencing (WGRS) has proven to be a powerful tool for genetic evaluation, selective sweep analysis and genetic relationship exploration. Huang et al. have uncovered the genetic structure and the molecular underpinnings of the SHCs trademark coloration using WGRS data (Huang et al. 2020b). Li et al. have explored the genetic signatures of high-altitude adaptation in Tibetan chickens by comparing the strong selection signatures genomic region of the Tibetan and lowland fowls (Li et al. 2019). Luo et al. performed a comparative genomics analysis for determining the behavioral pattern of gamecock chickens, and observed genetic introgression from commercial chickens into indigenous chickens (Luo et al. 2020).

2. M&M. line 68-70: the author provided some paraments for variant calls, but what is the basis for using these parameters?

Response: The paraments used in this article are the default parameters of GATK software, generally no need to modify. Other articles have the same usage, such as in Li (GigaScience, 2017); Deng (Current Biology, 2020).

3. Results. In line 114-117: The authors have divided the data into five categories, what is the significance of the existence of the categories? No subsequent analysis was seen.

Response: Thank you very much for your comments. We supplement and discuss this part in the Result and Discussion part in LINE 140-155 and LINE 295-300.

Isochores are long DNA fragments with uniform GC content, are tightly associated with many genomic biological characteristics such as recombination, GC3 content, and gene density; We divided into five categories (L1, L2, H1, H2 and H3) according to different GC levels (Gao et al. 2006; Li et al. 2009). 

4. Results. There are many changes to be made to the images. Fig3(A): The number of samples presented in the graph does not match the graph notes. There are many samples not shown in the figure notes. Fig3(B): BRA and DZAC colour distinction is not very good. It is difficult to see the difference between the two in the samples. Fig3(C): The result to be presented in the diagram is not clear. Three diagrams of similar samples in as close a colour match as possible

Response: Thank you very much for your correction. We have redrawn the diagram in the article and unified the color, sample number and corresponding notes.

5. Results. The principle of Fig. 4 is not made clear. What analysis is Figure 4 based on to obtain the results?

Response: According to the color of chicken skin, we divided the population into yellow skin groups (XYC, SHC, MA, DZAC and GDC) and non-yellow skin groups (LSFC, NDYC and WC).

6. Discussion. No good explanation of the results of the genetic diversity please revised it clearly and find for materials to enrich your discussion.

Response: Thank you very much for your suggestion. We have re-discuss by adding more materials. The content is in the Discussion in LINE 289-314.

Reviewer #2: My comments to authors are below (not in any particular order). The paper is brief, but very interesting, and deserves publication in Plos One after a few important corrections.

- I found the introduction very short and lacking information. For instance, it is very important for the reader to know what other efforts have been made to caracterize the Guangxi breeds? Have they ever been sequenced? Has similar work been done? What has not been done? What could justify the use of WGS over genotyping? Have the breeds been genotyped before? If they are in an official germplast catalog, they must have? Such studies are alluded to at lines 44 and 45 when "sporadic studies" are mentioned but we do not have any details. Please detail these studies, as this will make your study even more justified, which is important for the reader to contextualise your study with what has been done with these particular breeds, and also with other competing indigenous breeds in China. 

Response: Thank you very much for your suggestion and it is very helpful for us to sort out our ideas. We have reprogrammed the introduction, and the revised content is in LINE 47-63.

- Also in the introduction, sentences like "Indigenous chickens with delicious meat quality and unique flavors are more in line with the habits and preferences of domestic consumers" (lines 34-35) are highly subjective and a bit ideological. Has there been any study of choice preference or purchasing habits in this region to support this? I understand the need to breed from diverse stocks and especially from indigenous stocks, but it was not made apparent in this section.

Response: Thank you very much for your suggestion. Such a description is really not objective enough. We have added some materials in the Introduction for support in LINE 38-46.

- Regarding variant calling: did author estimate how much of the indigenous breed genomes did not map to the GRCg6a genome because of divergence? I understand it is difficult to proceed without reference genomes but you have sequenced the whole genomes. From an agricultural and breeding perspective, it would be quite interesting to know how much of the indigenous genomes is not represented by reference genomes from other breeds, and potentially how much "genetic novelty" there is in those genomes. For instance, it could be interesting to see this on Figure 1A (how much % instead of number of bases).

Response: Thank you very much for your advice and it is very helpful to improve the quality of our articles. We added the description of Novel SNPs to the results in LINE 134-142， Figure 1A shows the number of SNPs and indels. 

- Regarding figures, Figure 1A lacks appropriate Y-axis legends. Please correct. Generally speaking, all figures of this manuscript are quite low-quality (both in execution and in resolution). However, one thing that would be beneficial is to increase the font size. Many writings are not readable (especially in Figure 3A). It would be good to keep similar/related colour code for Figure 2D (I found myself struggling to remember the breed names I am unfamiliar with, perhaps worth providing the full names or whether they are Guangxi or not in the legend/color code too).

Response: Thank you very much. We have modified the table according to your comments and uploaded 330dpi resolution pictures on the submission website. The pictures in the article have unified colors and abbreviations.

- I am not a fan at all of the radial tree for the representation of diversity in Figure 3B. I also think the color code of this panel is very hard to read. It would be better to show a rectangular tree, rooted on red-jungle fowls and annotated clearly with all the breeds rather than showing it like this.

Response: Thank you very much. We adopted your suggestion and redrew a rectangular tree with RJF as the root.

- I suspect WL is White Leghorn, and not White Layers. Please correct in the manuscript where applicable.

Response: We are sorry for this mistake. We have carefully corrected White Leghorn throughout the manuscript according to your comment. 

- Please annotate Figure 4 with the loci of interest and thresholds to consider an interesting locus or not. It would be much more useful than the current version.

Response: Thank you very much. We have modified Figure 4 according to your suggestion.

---

## [Decision Letter · Decision Letter 1]

24 Dec 2021

PONE-D-21-10366R1Whole genome sequencing revealed genetic diversity and selection of Guangxi indigenous chickensPLOS ONE

Dear Dr. Yang,

Thank you for submitting your manuscript to PLOS ONE. After careful consideration, we feel that it has merit but does not fully meet PLOS ONE’s publication criteria as it currently stands. Therefore, we invite you to submit a revised version of the manuscript that addresses the points raised during the review process.

We look forward to receiving your revised manuscript.

Kind regards,

Tzen-Yuh Chiang

Academic Editor

PLOS ONE

Journal Requirements:

**Staff Editor Comments**

1. For editorial purposes please include a legend on the Y-axis for figure 1A. The legend text indicates that this represents SNP and Indel number, but the scale goes from 0-0.16.

Reviewers' comments:

Reviewer's Responses to Questions

**Comments to the Author**

1. If the authors have adequately addressed your comments raised in a previous round of review and you feel that this manuscript is now acceptable for publication, you may indicate that here to bypass the “Comments to the Author” section, enter your conflict of interest statement in the “Confidential to Editor” section, and submit your "Accept" recommendation.

Reviewer #2: (No Response)

2. Is the manuscript technically sound, and do the data support the conclusions?

Reviewer #2: Yes

3. Has the statistical analysis been performed appropriately and rigorously? 

Reviewer #2: Yes

4. Have the authors made all data underlying the findings in their manuscript fully available?

Reviewer #2: Yes

5. Is the manuscript presented in an intelligible fashion and written in standard English?

Reviewer #2: Yes

6. Review Comments to the Author

Reviewer #2: (No Response)

7. PLOS authors have the option to publish the peer review history of their article (what does this mean?). If published, this will include your full peer review and any attached files.

Reviewer #2: No

---

## [Author Response · Author response to Decision Letter 1]

13 Jan 2022

Thank you your valuable advice. We have modified figure 1A with a Y-axis legend renamed ‘PACE-FIG 1’ on the submission website. 

We re-examined the references list to make sure that the information was complete and correct and no papers have been retracted.

The PACE corrected figures has been uploaded to the submission website and renamed as PACE-FIG1~4.

---

## [Decision Letter · Decision Letter 2]

1 Mar 2022

Whole genome sequencing revealed genetic diversity and selection of Guangxi indigenous chickens

PONE-D-21-10366R2

Dear Dr. Yang,

We’re pleased to inform you that your manuscript has been judged scientifically suitable for publication and will be formally accepted for publication once it meets all outstanding technical requirements.

Kind regards,

Tzen-Yuh Chiang

Academic Editor

PLOS ONE

Additional Editor Comments (optional):

Reviewers' comments:

Reviewer's Responses to Questions

**Comments to the Author**

1. If the authors have adequately addressed your comments raised in a previous round of review and you feel that this manuscript is now acceptable for publication, you may indicate that here to bypass the “Comments to the Author” section, enter your conflict of interest statement in the “Confidential to Editor” section, and submit your "Accept" recommendation.

Reviewer #2: All comments have been addressed

2. Is the manuscript technically sound, and do the data support the conclusions?

Reviewer #2: Yes

3. Has the statistical analysis been performed appropriately and rigorously? 

Reviewer #2: Yes

4. Have the authors made all data underlying the findings in their manuscript fully available?

Reviewer #2: (No Response)

5. Is the manuscript presented in an intelligible fashion and written in standard English?

Reviewer #2: Yes

6. Review Comments to the Author

Reviewer #2: (No Response)

7. PLOS authors have the option to publish the peer review history of their article (what does this mean?). If published, this will include your full peer review and any attached files.

Reviewer #2: No

---

## [Editor Report · Acceptance letter]

7 Mar 2022

PONE-D-21-10366R2 

Whole-genome sequencing revealed genetic diversity and selection of Guangxi indigenous chickens 

Dear Dr. Yang:

I'm pleased to inform you that your manuscript has been deemed suitable for publication in PLOS ONE. Congratulations! Your manuscript is now with our production department. 

Kind regards, 

on behalf of

Dr. Tzen-Yuh Chiang 

Academic Editor

PLOS ONE